# ICU-Acquired Colonization and Infection Related to Multidrug-Resistant Bacteria in COVID-19 Patients: A Narrative Review

**DOI:** 10.3390/antibiotics12091464

**Published:** 2023-09-20

**Authors:** Alexandre Gaudet, Louis Kreitmann, Saad Nseir

**Affiliations:** 1Médecine Intensive Réanimation, CHU de Lille, F-59000 Lille, France; alexandre.gaudet@chu-lille.fr; 2CNRS, Inserm U1019-UMR9017-CIIL-Centre d’Infection et d’Immunité de Lille, Institut Pasteur de Lille, CHU Lille, Université de Lille, F-59000 Lille, France; 3Centre for Antimicrobial Optimisation, Department of Infectious Disease, Faculty of Medicine, Imperial College London, London W12 0HS, UK; louis.kreitmann@gmail.com; 4Department of Intensive Care Medicine, Imperial College Healthcare NHS Trust, London NW1 5QH, UK; 5Inserm U1285, Université de Lille, CNRS, UMR 8576-UGSF, F-59000 Lille, France

**Keywords:** antimicrobial resistance, multidrug resistance, cross-infection, ventilator-associated pneumonia, COVID-19, bloodstream infection, critical illness, intensive care units

## Abstract

A large proportion of ICU-acquired infections are related to multidrug-resistant bacteria (MDR). Infections caused by these bacteria are associated with increased mortality, and prolonged duration of mechanical ventilation and ICU stay. The aim of this narrative review is to report on the association between COVID-19 and ICU-acquired colonization or infection related to MDR bacteria. Although a huge amount of literature is available on COVID-19 and MDR bacteria, only a few clinical trials have properly evaluated the association between them using a non-COVID-19 control group and accurate design and statistical methods. The results of these studies suggest that COVID-19 patients are at a similar risk of ICU-acquired MDR colonization compared to non-COVID-19 controls. However, a higher risk of ICU-acquired infection related to MDR bacteria has been reported in several studies, mainly ventilator-associated pneumonia and bloodstream infection. Several potential explanations could be provided for the high incidence of ICU-acquired infections related to MDR. Immunomodulatory treatments, such as corticosteroids, JAK2 inhibitors, and IL-6 receptor antagonist, might play a role in the pathogenesis of these infections. Additionally, a longer stay in the ICU was reported in COVID-19 patients, resulting in higher exposure to well-known risk factors for ICU-acquired MDR infections, such as invasive procedures and antimicrobial treatment. Another possible explanation is the surge during successive COVID-19 waves, with excessive workload and low compliance with preventive measures. Further studies should evaluate the evolution of the incidence of ICU-acquired infections related to MDR bacteria, given the change in COVID-19 patient profiles. A better understanding of the immune status of critically ill COVID-19 patients is required to move to personalized treatment and reduce the risk of ICU-acquired infections. The role of specific preventive measures, such as targeted immunomodulation, should be investigated.

## 1. Introduction

Patients affected by the most severe form of coronavirus disease 2019 (COVID-19) pneumonia develop acute respiratory distress syndrome (ARDS), characterized by profound and life-threatening hypoxemia. These patients often have a prolonged intensive care unit (ICU) stay and require an extended duration of invasive mechanical ventilation (IMV) [1]. Several studies have shown that critically ill COVID-19 patients present a higher incidence of ICU-acquired infections than non-COVID-19 controls, especially hospital- and ventilator-associated pneumonia (HAP and VAP, respectively) [2] and bloodstream infections (BSI) [3]. They are exposed to broad-spectrum antimicrobials [4], which leads to a sustained antibiotic selection pressure. Consequently, there have been major concerns about the potential impact of the COVID-19 pandemic on the emergence and spread of antimicrobial resistance (AMR) [5].

AMR is an important concern in ICUs, where a significant proportion of secondary infections are attributed to multidrug-resistant (MDR) bacteria [6]. In studies conducted prior to the pandemic, ICU-acquired colonization with MDR bacteria was linked to a longer ICU length-of-stay [7], and ICU-acquired infection with MDR bacteria was associated with a longer duration of IMV [8] and higher mortality [6,9].

Several studies have investigated the burden of AMR among COVID-19 patients (see reviews in [10,11,12,13,14,15]), but most presented important methodological limitations. These include a limited sample size, a retrospective and/or monocentric design, the inclusion of both ICU and non-ICU patients, the absence of a control group of non-COVID-19 patients, and the failure to use appropriate statistical methods to account for important confounding factors.

The main objective of this narrative review is to summarize the evidence from studies that have appropriately investigated the association between COVID-19 and the incidence of ICU-acquired infection and colonization with MDR bacteria. The secondary objectives were to describe the pathophysiology and risk factors for ICU-acquired infection and colonization with MDR bacteria among critically ill COVID-19 patients, to investigate the impact of COVID-19 on the association between ICU-acquired infection and colonization with MDR bacteria and prognostic outcomes, and to review specific preventive measures and future lines of research to limit the emergence and spread of AMR in this population (Figure 1).

## 2. Methods

We searched the MEDLINE and PubMed databases for articles in English related to bacterial co-infections, secondary infections, bacterial colonization, and/or antibiotic resistance among COVID-19 patients. Further references were added through hand-searching in the relevant literature and verifying references of key papers. We screened the titles and abstracts of papers identified by our search, and assessed the full text of potentially relevant articles. The inclusion of papers in the final manuscript was based on consensus among all three co-authors.

To investigate the primary endpoint (i.e., association between COVID-19 and the incidence of ICU-acquired infection and colonization with MDR bacteria), we selected articles with the following criteria: (1) they provided original data on antibiotic resistance in bacteria isolated from COVID-19 patients; (2) they provided data on ICU-acquired infections and/or colonization with resistant bacteria (i.e., bacteria isolated >48 h following ICU admission); (3) they included a control group of non-COVID-19 patients; and (4) they provided a formal statistical assessment of the relationship between COVID-19 status and any quantitative estimate of antibiotic resistance (e.g., incidence of ICU-acquired infection with multidrug-resistant bacteria, or frequency of resistance to a given antibiotic among bacteria isolated from a given body site), or they provided enough data to perform such assessment.

We assessed the characteristics of the selected studies, including: (1) the study design, setting and sample size (Table 1); (2) whether they presented data on ICU-acquired colonization with resistant bacteria, and if so, whether this had been assessed through systematic measurement (e.g., nasal and/or rectal swabs); (3) whether they presented data on ICU-acquired infection with resistant bacteria, and if so, which type of infection (e.g., VAP, BSI, or all), and whether they also presented data on ICU-acquired infection related to non-resistant bacteria; (4) whether the study included patient-related data, and whether statistical analyses had been adjusted for patient-related factors known to impact the incidence of ICU-acquired infection/colonization with resistant bacteria (Table 2); (5) the main findings regarding the association of COVID-19 with the incidence of ICU-acquired MDR colonization/infection; (6) the main findings regarding the risk factors for ICU-acquired MDR colonization/infection among COVID-19 ICU patients; (7) the main findings regarding the effect of COVID-19 on the association between ICU-acquired MDR colonization/infection and prognostic outcomes (ICU length-of-stay (LOS), duration of IMV, and mortality); and (8) the main methodological limitations (Table 3).

To investigate secondary endpoints, we additionally considered articles related to the pathophysiology and preventive measures of ICU-acquired colonization and infection related to MDR bacteria among critically ill COVID-19 patients.

## 3. Epidemiology

As presented in Table 1, 24 studies providing estimates of the association between COVID-19 status and ICU-acquired infection/colonization with resistant bacteria fulfilled our inclusion criteria [2,16,17,18,19,20,21,22,23,24,25,26,27,28,29,30,31,32,33,34,35,36,37,38]. 

The majority of these studies presented important methodological limitations, mainly related to a retrospective (*n* = 21) or single-centered (*n* = 15) design. Most studies compared COVID-19 ICU patients with a control group of non-COVID-19 ICU patients enrolled before the onset of the pandemic. Most studies (*n* = 22) focused on ICU-acquired infections, and eight also evaluated ICU-acquired colonization with resistant bacteria. The assessment of antibiotic resistance relied on classical microbiology criteria, but the definitions used to classify bacteria as “difficult-to-treat” or “multidrug-resistant” varied across studies.

Besides the major risk of bias related to the above-mentioned methodological limitations, the most important challenge we faced when attempting to summarize the evidence of the association between COVID-19 and antibiotic resistance in the setting of ICU lay in the variability in the assessment of antibiotic resistance across studies. As presented in Table 2, some studies have estimated the incidence of ICU-acquired infection/colonization with MDR bacteria, while others have documented the incidence of infection/colonization with bacteria of all resistance profiles along with the rates (or frequencies) of resistant strains among isolated bacteria, which made comparison across studies extremely challenging. 

Finally, several studies have mostly reported bacteriological data and have failed to provide a clear link between bacteriology (e.g., the result of a blood or tracheal aspirate culture) and patient-related data (infection vs. colonization status, clinical variables, etc.), making any attempt to precisely evaluate the incidence, risk factors, and prognostic impact of ICU-acquired MDR colonization/infection impossible. In line with this, only eight studies have adjusted statistical analyses on patient-level factors known to be associated with the occurrence of ICU-acquired infection/colonization with resistant bacteria.

The main findings of the selected studies on the association between COVID-19 and the incidence of ICU-acquired colonization and infection with MDR bacteria are summarized in Table 3. In the COVID-BMR study, a prospective multicenter study in seven ICUs comparing 367 COVID-19 patients with 680 controls recruited in the same centers before the pandemic, we found that COVID-19 patients had a higher cumulative incidence of ICU-acquired infection with MDR bacteria, both prior to and after adjustment for baseline confounders (adjusted sub-distribution hazard ratio (cHR) 2.50, 95% CI 1.90–3.28) [37]. In this study, the main types of ICU-acquired infections related to MDR bacteria were HAP, VAP, and BSI.

Regarding the incidence of VAP related to MDR bacteria, in the COVAPID study, a retrospective multicenter study (36 ICUs) comparing 568 COVID-19 patients with 1008 controls, the incidence of VAP was significantly higher in COVID-19 patients (36.1%) than in patients with flu (22.2%) or no viral infection (16.5%), but the frequency of MDR strains was lower in COVID-19 patients (23.3%) than in patients with flu (38.4%) and no viral infection (33.8%) [2,39]. Data from this paper can be used to provide estimates of the cumulative incidence of VAP related to MDR bacteria in all three groups (11.8% in COVID-19 patients, 11.6% in flu patients, and 8.6% in patients with no viral infection), which does not suggest that this incidence might be higher among COVID-19 patients (contrarily to the COVID-BMR study [37]). However, in a retrospective multicenter study on 1879 COVID-19 patients and 1879 controls in 94 ICUs, the incidence of a first episode of VAP was higher in COVID-19 patients (adjusted sHR 1.68, 95% CI 1.45–1.96), with a similar frequency of MDR pathogens (except for a lower frequency of MRSA), which would suggest a higher incidence of VAP episodes related to MDR bacteria in the COVID-19 group [24] (in line with the COVID-BMR study [37]). Similarly, in a retrospective monocentric study, COVID-19 patients (*n* = 90) had a higher incidence of VAP (sHR 1.72, 95% CI 1.14–2.57), including MDR VAP (23% vs. 11%, *p* = 0.03), than non-COVID-19 controls (*n* = 82; in line with the COVID-BMR study [37]). Finally, using the REA-REZO surveillance network database, a large retrospective study in France found a higher incidence of VAP [26] among 1687 COVID-19 patients than controls (72,258 non-COVID-19 ICU patients recruited before and during the pandemic), with similar proportions of resistant strains among isolated bacteria, again suggesting that the incidence of ICU-acquired infections related to MDR bacteria might be higher among COVID-19 patients. In conclusion, even if no study has been designed to specifically assess this endpoint, the existing literature suggests that COVID-19 patients may have a higher incidence of VAP related to MDR bacteria than controls, mostly linked to a higher incidence of VAP (related to non-MDR and MDR strains altogether) [13].

Regarding the incidence of BSI related to MDR bacteria, a retrospective monocentric study on 497 COVID-19 patients and 823 controls reported that COVID-19 patients had a higher incidence of BSI related to MDR bacteria [38] (adjusted cause-specific HR (cHR) 2.65, 95% CI 1.25–5.59, in line with the COVID-BMR study [37]). Similarly, a retrospective international study in 53 ICUs found an increased incidence of hospital-acquired BSI related to difficult-to-treat Gram-negative bacteria in COVID-19 (*n* = 252) vs. controls (*n* = 577; 19.4% vs. 13%, *p* = 0.017) [29]. Finally, a large retrospective study in France found a higher incidence of VAP and BSI related to MDR bacteria in COVID-19 patients [30]. In conclusion, the existing literature suggests that COVID-19 patients may have a higher incidence of BSI related to MDR bacteria than controls.

The occurrence of ICU-acquired infections with MDR bacteria is influenced by several factors, but is preceded by colonization in the majority of cases. In the COVID-BMR study, we found that there was no significant difference in the cumulative incidence of ICU-acquired MDR colonization between COVID-19 patients and controls (34.1% vs. 27.9%, adjusted sHR 1.27, 95% CI 0.85–1.88) [37]. These estimates were in line with a monocentric retrospective study where the cumulative incidence of ICU-acquired colonization with MDR bacteria was not statistically different in COVID-19 patients when compared with matched controls (respectively 33% vs. 21% sHR 1.71, 95% CI 0.93–3.12) [16]. Among the studies included in our review, two monocentric retrospective studies reported a similar incidence of ICU-acquired colonization with MDR bacteria in COVID-19 patients vs. controls [18,20]; a monocentric retrospective found a lower incidence of ICU-acquired MDR colonization in COVID-19 vs. control patients (47.4% vs. 81.4%, *p* = 0.005) [27]; and a multicenter retrospective study found decreased rates of MRSA, VRE, CRE and CRAB in systematic screening samples from COVID-19 in comparison with non-COVID-19 patients [25]. In conclusion, the existing literature suggests that the incidence of ICU-acquired colonization with MDR bacteria is similar or potentially lower in COVID-19 patients vs. controls.

As we discussed in the COVID-BMR study, assessing ICU-acquired MDR colonization and infection separately enables a more precise investigation of the mechanisms leading to the emergence and spread of AMR among COVID-19 ICU patients [37]. As ICU-acquired MDR colonization could be related to the cross-transmission of MDR strains across patients through direct contact with healthcare workers, our findings would suggest that organizational changes triggered by the COVID-19 pandemic—such as the cohorting of patients in dedicated COVID-19 units with strict enforcement of infection prevention and control (IPC) policies, enhanced hand hygiene, use of protective personal equipment (PPE), and efforts to decrease patient contacts—might have had a positive impact on this phenomenon. Second, our findings also suggest that COVID-19 patients might harbor distinct characteristics that make them more susceptible than controls to developing ICU-acquired MDR infections once colonized with a given MDR strain; these factors will be discussed in more detail in paragraph 4 related to pathophysiology.

Regarding the impact of COVID-19 status on the association between ICU-acquired MDR colonization/infection and prognostic outcomes, we have documented, in the COVID-BMR study, that the occurrence of a combined outcome including ICU-acquired colonization and/or infection with MDR bacteria was associated with decreased survival (adjusted cHR 2.61, 95% CI 1.59–4.27) in COVID-19 patients, but not in controls [37]. Interestingly, similar findings were obtained in the study by Buetti et al. (looking at BSI related to DTR Gram-negative bacteria) [29] and in the study by Cogliati Dezza et al. (looking at BSI related to MDR Gram-negative bacteria) [27]. In the study by Piantoni et al., we found that ICU-acquired MDR BSI was associated with increased mortality in the overall cohort (adjusted HR 1.73, 95% CI 1.0–3.0), with no effect of COVID-19 status on this association [38]. Both in COVID-BMR [37] and in the study by Piantoni et al. [38], there was no impact of the occurrence of ICU-acquired colonization and/or infection with MDR bacteria on ICU LOS and on the duration of IMV in the overall cohort, in COVID-19 patients, and in controls.

As presented in Table 3, very few studies have looked into the specific risk factors for ICU-acquired colonization/infection among COVID-19 patients.

## 4. Pathophysiology

Recent epidemiological data have brought to light a notable increase in the risk of MDR infections among patients hospitalized in ICUs with severe COVID-19. This increased susceptibility to MDR infections can be attributed to several factors. Primarily, the administration of immunomodulatory treatments in the management of critically ill COVID-19 patients in ICUs plays a significant role. These treatments encompass various modalities, including, but not limited to, the widespread use of dexamethasone along with potential adjunctive therapies, such as JAK2 inhibitors and IL-6 receptor antagonists. Thus, in the COVID-BMR study, treatment with steroids was found in 75% of COVID-19 patients, vs. 22% of non-COVID-19 patients [37]. The immunosuppressive effects resulting from these treatments may be contributory factors for the elevated incidence of infections, particularly MDR infections. Interestingly, according to findings from the COVID-BMR study, the increased risk of MDR infection among COVID-19 patients occurs as early as seven days following admission to the ICU [37]. Conversely, other data have identified a heightened risk of VAP occurring around three weeks after admission to the ICU in patients exposed to dexamethasone [40,41]. This apparent discrepancy in the timings of infection onset seems to indicate that other factors are likely to contribute to the higher incidence of MDR infections in critically ill COVID-19 patients.

Consistent data have shed light on the direct involvement of the SARS-CoV-2 virus in triggering post-aggressive immunoparalysis. Comparatively, the inflammatory response to COVID-19 appears to be generally of a more moderate intensity when compared with sepsis, with a prolonged state of immunoparalysis observed among severely ill COVID-19 patients. However, it is crucial to note that immune dysregulation in COVID-19 patients exhibits a heterogeneous pattern. Notably, there is a high incidence of lymphopenia affecting both B and T cell populations, with subsequent impacts on cellular and humoral responses. Furthermore, studies have also indicated the presence of monocyte dysfunction, whose intensity seemingly correlates with disease severity [42,43,44,45]. These immune alterations collectively contribute to the development of an immunodeficient state intrinsically linked to COVID-19, thereby rendering severe COVID-19 patients more susceptible to MDR infections.

Another potential explanation for the higher risk of MDR infections among severely ill COVID-19 patients is the increased likelihood of MDR emergence due to the selection pressure resulting from antibiotic use. Notably, exposure to antibiotics stands as one of the primary risk factors for the development of MDR in ICUs [46]. Furthermore, compelling evidence suggests an elevated frequency of antibiotic administration among severely ill COVID-19 patients, potentially resulting from relaxed antibiotic stewardship during the pandemic. Data from the ISARIC consortium, encompassing nearly 50,000 hospitalized COVID-19 patients, revealed that 37% of patients had been exposed to antibiotics prior to hospital admission. This exposure frequency further increased to over 85% during the initial phase of hospitalization, despite bacterial sampling being conducted in only 17% of patients, with positive bacterial cultures found in less than 3% of the total cohort [4,47]. Consequently, the low occurrence of bacterial coinfections is consistent with the findings from the extensive European study COVID-ICU, which reported a 7% frequency of bacterial coinfections upon admission to the ICU [1]. Similarly, a recent meta-analysis involving over 30,000 patients reported a frequency of confirmed bacterial coinfection during the initial phase of approximately 4%, while the rate of antibiotic administration was 60% [48]. These data highlight the significant disparity between the low occurrence of initial bacterial coinfections in severely ill hospitalized COVID-19 patients and the high frequency of antibiotic exposure.

Furthermore, other hypotheses regarding the characteristics of hospital stays can be postulated to explain the increased risk of MDR infections among patients admitted to the ICU for COVID-19. According to data from the COVID-BMR study, the median length of stay in the ICU for severely ill COVID-19 patients was found to be 15 days, compared with 10 days for patients admitted for other reasons [37]. Additionally, the authors of this study noted increased exposure to invasive medical devices among COVID-19 patients compared with non-COVID-19 patients. This included a higher utilization of extracorporeal membrane oxygenation (ECMO), observed in over 9% of COVID-19 patients compared with 2% of non-COVID-19 patients, as well as arterial catheters, utilized in 97% of COVID-19 patients versus 83% of non-COVID-19 patients. Notably, the study also highlighted an extended duration of exposure to these invasive devices. As compared with non-COVID-19 cases, COVID-19 patients exhibited a median duration of exposure of 16 versus 11 days for central venous catheters, 14 versus 9 days for arterial catheters, and 15 versus 8 days for invasive mechanical ventilation. Consequently, the increased frequency of breaches in the anatomical barrier resulting from prolonged exposure to invasive devices likely contributed to the increased risk of infection among MDR-colonized patients, thereby amplifying the incidence of MDR infections among COVID-19 patients in ICUs.

Lastly, material and organizational factors, such as a surge in patient admissions to ICUs, may also participate in the increased risk of MDR infections among patients admitted for COVID-19. The relationship between the scarcity of ICU beds and the prognosis of hospitalized COVID-19 patients has been well-documented since the initial waves of the pandemic [49,50]. Several factors have been identified to explain this relationship, including a shortage of nursing staff leading to an excessive workload [51], which, in turn, may result in reduced adherence to individual protective measures. Similarly, the inadequate availability of personal protective equipment has also been identified as a contributing factor during the pandemic, potentially heightening the risk of MDR transmission [52].

## 5. Specific Preventive Measures

Several studies reported a higher incidence of ICU-acquired infections, including those related to MDR bacteria in COVID-19 patients [2,37]. Although this high incidence could be explained by exposure to well-known risk factors, such as invasive procedures and long ICU stays, the role of compliance with preventive measures during the surge was not evaluated. One could argue that general preventive measures should be strictly applied in COVID-19 patients, given the high incidence of ICU-acquired infections in these patients.

To our knowledge, no randomized controlled study has evaluated the efficiency of measures aiming to prevent ICU-acquired infection in COVID-19 patients. Previous studies highlighted a link between microbiota alteration in COVID-19 patients and the severity of inflammation, persistence of ARDS, and mortality [53,54,55,56]. The role of digestive and pulmonary microbiota in the pathogenesis of ICU-acquired infections has also been suggested [57,58]. However, whether the alteration of microbiota results in different effects on ICU-acquired infection in COVID-19 patients, as compared with non-COVID-19 patients is still to be evaluated. In a recent pilot study performed in 17 COVID-19 patients, gut microbiota composition significantly differed between patients with ICU-acquired colonization related to MDR bacteria, compared with those with no MDR bacteria colonization [59]. Further studies are needed to confirm these findings.

## 6. Future Research

In light of these data, several research paths could be developed in the future to better understand the relationship between COVID-19 and MDR infections.

Firstly, we can question the evolution of the profile of COVID-19 patients. Epidemiological data highlight an increasing proportion of immunocompromised individuals [60] among severely ill COVID-19 patients. Interestingly, recent data suggest a decreased risk of colonization and/or infection with MDR in immunocompromised COVID-19 patients compared with immunocompetent subjects [61]. Thus, the increased proportion of immunocompromised patients among COVID-19 patients may actually decrease the risk of MDR infection in this population.

As previously mentioned, the frequency of immunomodulatory therapies appears to be involved in the increased risk of MDR infection among COVID-19 patients. Immunological investigations highlight significant variability in the degree of pulmonary inflammatory responses among these patients, which seems to correlate with disease severity [62]. These data support the potential benefit of a personalized approach, aiming to initiate immunomodulatory treatments in the most inflammatory patients. Conversely, this strategy could spare immunomodulatory treatments for the least inflammatory patients, for whom the risk–benefit ratio of these therapies may be unfavorable. Studies aiming to explore the potential benefit of such a strategy could thus help to better target the use of immunomodulators in this population.

Finally, other research directions also seem interesting to explore with the goal of reducing the incidence of MDR infections in severely ill COVID-19 patients. These include the use of innovative real-time machine-learning tools incorporating dynamic patient-contact networks to predict colonization and infection with MDR bacteria [63,64]; novel point-of-care molecular diagnostic tests to detect co-infections and decrease the inappropriate use of antibiotics [65]; or the use of specific bundles of care to prevent the incidence of specific ICU-acquired infections [66].

## 7. Conclusions

Owing to the potential impact of the COVID-19 pandemic on antibiotic stewardship programs and the correct application of IPC measures in hospitals, there have been major concerns about the consequences of COVID-19 on AMR. In this narrative review, we report that critically ill COVID-19 patients, compared with controls, have been shown to present a higher incidence of ICU-acquired infections—particularly VAP and BSI—related to MDR bacteria, despite a similar incidence of ICU-acquired colonization with MDR bacteria. Several pathophysiological factors can explain these findings, including SARS-CoV-2-mediated post-aggressive immunoparalysis, different exposure to antibiotics, steroids, and other immunomodulating agents, a longer ICU LOS and exposure to invasive devices in COVID-19 patients, and organizational constraints triggered by the pandemic. Future lines of research notably include precision medicine approaches to immune modulation in ICU patients.

## Figures and Tables

**Figure 1 antibiotics-12-01464-f001:**
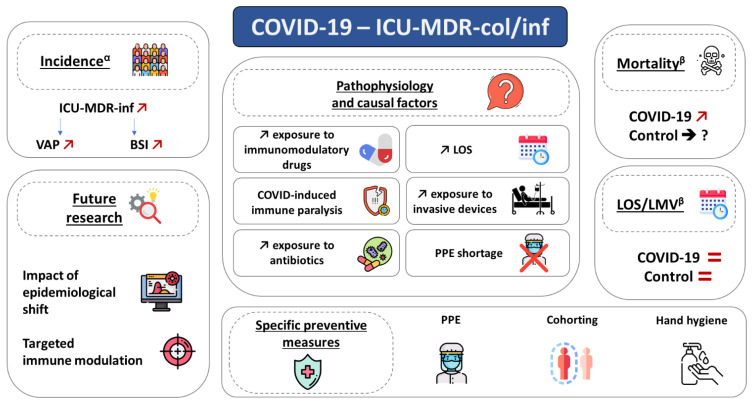
ICU-acquired colonization and infection related to multidrug-resistant bacteria in COVID-19 patients: incidence, outcomes, pathophysiology, prevention, and future research axis. BSI, bloodstream infection; ICU-MDR-col/inf, Intensive care unit-acquired multidrug-resistant bacteria colonization or infection; LMV, length of mechanical ventilation; LOS, length of stay; PPE, personal protection equipment; VAP, ventilator-associated pneumonia. α: in COVID-19 vs. non-COVID-19. β: in MDR vs. no MDR.

**Table 1 antibiotics-12-01464-t001:** Characteristics of the studies included in the review for the assessment of the association between COVID-19 and the incidence of ICU-acquired colonization and infection related to MDR bacteria (primary endpoint).

Study	Year	Journal	Setting	Nb. of Centers	Design	Sample Size (Cases vs. Controls)
Bogossian et al. [16]	November 2020	*Microorganisms*	Belgium	1	Retrospective	72/72
Razazi et al. [17]	December 2020	*Critical Care*	France	1	Retrospective	90/82
Oliva et al. [18]	January 2021	*Infection*	Italy	1	Retrospective	55/19
Rouzé et al. [2]	January 2021	*Intensive Care Medicine*	Europe	36	Retrospective	568/1008
Ong et al. [19]	August 2021	*Antimicrobial Resistance and Infection Control*	Singapore	1	Prospective	71/487
Rouyer et al. [20]	August 2021	*Antibiotics*	France	1	Retrospective	79/188
Zuglian et al. [21]	February 2022	*BMC Infectious Diseases*	Italy	1	Retrospective	176/194
Bahçe et al. [22]	March 2022	*Microbial Pathogenesis*	Turkey	1	Retrospective	602/971
Sathyakamala et al. [23]	March 2022	*Journal of Preventive Medicine and Hygiene*	India	1	Retrospective	356/292
Vacheron et al. [24]	March 2022	*Critical Care Medicine*	France	94	Retrospective	1879/1879
Jeon et al. [25]	April 2022	*Antibiotics*	Korea	4	Retrospective	209,107 (total including ICU and non-ICU patients, pre- and per-COVID-19 periods)
Vacheron et al. [26]	July 2022	*American Journal of Respiratory and Critical Care Medicine*	France	?	Retrospective	1687/72,258
Cogliati Dezza et al. [27]	July 2022	*Antibiotics*	Italy	1	Retrospective	18/28
Metan et al. [28]	August 2022	*GMS Hygiene and Infection Control*	Turkey	1	Retrospective	?
Buetti et al. [29]	October 2022	*Critical Care*	World	53	Retrospective	252/577
Lepape et al. [30]	October 2022	*Clinical Microbiology and Infection*	France	110	Retrospective	4465/63,433
Kinross et al. [31]	November 2022	*Eurosurveillance*	Europe	?	Retrospective	?
Segala et al. [32]	March 2023	*Infection*	Italy	1	Prospective	14,884 (total including cases and controls)
Önal et al. [33]	April 2023	*Le Infezioni in Medicina*	Turkey	1	Retrospective	8157 (total including cases and controls)
Petrakis et al. [34]	May 2023	*Pathogens*	Greece	1	Retrospective	823/393
Lee et al. [35]	June 2023	*Journal of Hospital Infection*	Korea	346	Retrospective	?
Chang et al. [36]	July 2023	*Journal of the Formosan Medical Association*	Taiwan	1	Retrospective	38,184 ICU patients (total)
Kreitmann et al. [37]	July 2023	*Intensive Care Medicine*	France	7	Prospective	367/680
Piantoni et al. [38]	July 2023	*Antibiotics*	France	1	Retrospective	497/823

ICU, intensive care unit. ? indicates no data in original manuscript.

**Table 2 antibiotics-12-01464-t002:** Variables reported in the studies included in the review for the assessment of the primary endpoint.

Study	Data on ICU Patients	Control Group	Assessment of ICU-Acquired MDR Colonization	Systematic Screening of ICU-Acquired MDR Colonization	Assessment of ICU-Acquired MDR Infections	Reporting of ICU-Acquired Non-MDR Infections	Adjustment for Confounding Factors
Bogossian et al. [16]	Yes	Non-COVID-19 ICU patients before pandemic	Yes	Yes	No	No	Yes
Razazi et al. [17]	Yes	Non-COVID-19 ICU patients before and during pandemic	No	No	Yes (VAP)	Yes	Yes
Oliva et al. [18]	Yes	Flu patients (before pandemic)	Yes	Yes	Yes (all)	Yes	No
Rouzé et al. [2]	Yes	ICU patients with flu or no viral infection (before and during pandemic)	No	No	Yes (VA-LTRI, VAP and VAT)	Yes	Yes
Ong et al. [19]	Yes	Non-COVID-19 ICU patients during pandemic	No	No	Yes (all)	Yes	Yes
Rouyer et al. [20]	Yes	Non-COVID-19 ICU patients during pandemic	Yes	Yes	Yes (VAP)	Yes	No
Zuglian et al. [21]	Yes	Non-COVID-19 IUC patients before pandemic	No	No	Yes (respiratory samples)	Yes	No
Bahçe et al. [22]	Yes	Non-COVID-19 ICU patients before pandemic	No	No	Yes (respiratory samples)	Yes	No
Sathyakamala et al. [23]	Yes	Non-COVID-19 ICU patients during pandemic	No	No	Yes (all)	Yes	No
Vacheron et al. [24]	Yes	Non-COVID-19 ICU patients before pandemic	No	No	Yes (VAP)	Yes	Yes
Jeon et al. [25]	Yes	Non-COVID-19 ICU patients before pandemic	Yes	?	Yes (all)	No	No
Vacheron et al. [26]	Yes	Non-COVID-19 ICU patients before and during pandemic	No	No	Yes (VAP)	Yes	No
Cogliati Dezza et al. [27]	Yes	Non-COVID-19 ICU patients before pandemic	Yes	Yes	Yes (MDR Gram-negative BSI)	No	No
Metan et al. [28]	Yes	Non-COVID-19 ICU patients during pandemic	No	No	Yes (BSI)	No	No
Buetti et al. [29]	Yes	Non-COVID-19 ICU patients before pandemic	No	No	Yes (BSI)	Yes	No
Lepape et al. [30]	Yes	Non-COVID-19 ICU patients during and before pandemic	Yes	?	Yes (VAP, HAP, BSI)	Yes	Yes
Kinross et al. [31]	Yes	Non-COVID-19 ICU patients during and before pandemic	No	No	Yes (*Acinetobacter* spp. BSI)	Yes	No
Segala et al. [32]	Yes	Non-COVID-19 ICU patients before pandemic	Yes	?	Yes (BSI)	Yes	No
Önal et al. [33]	Yes	Non-COVID-19 ICU patients before and during pandemic	No	No	Yes (all)	Yes	No
Petrakis et al. [34]	Yes	Non-COVID-19 ICU patients during pandemic	No	No	Yes (blood and respiratory samples)	Yes	No
Lee et al. [35]	Yes	Non-COVID-19 ICU patients before pandemic	No	No	Yes (all)	Yes	No
Chang et al. [36]	Yes	Non-COVID-19 ICU patients before pandemic	No	No	Yes (all)	No	No
Kreitmann et al. [37]	Yes	Non-COVID-19 ICU patients before pandemic	Yes	Yes	Yes (all)	No	Yes
Piantoni et al. [38]	Yes	Non-COVID-19 ICU patients during same period of follow-up	No	No	Yes (ICU-acquired MDR BSI)	No	Yes

BSI, bloodstream infection; HAP, hospital-associated infection; ICU, intensive care unit; MDR, multidrug-resistant; VAP, ventilator-associated pneumonia; VA-LRTI, ventilator-associated lower respiratory tract infection. ? indicates no data in original manuscript.

**Table 3 antibiotics-12-01464-t003:** Main findings from the studies included in the review for the assessment of the primary endpoint.

Study	Incidence of ICU-Acquired MDR Colonization and Infection	Risk Factors for ICU-Acquired Colonization or Infection among COVID-19 Patients	Impact on Outcomes	Limitations
Bogossian et al. [16]	No significant association between COVID-19 status and the incidence of ICU-acquired MDR colonization (sHR 1.71 (CI 95% 0.93–3.21)	Risk factors for ICU-acquired MDR colonization: vasopressors, antimicrobial therapy	Longer duration of ICU and hospital LOS, but no impact on mortality in patients with ICU-acquired MDR colonization (among COVID-19 patients)	Small sample size, monocentric study
Razazi et al. [17]	COVID-19 patients had higher incidence of VAP (sHR 1.72, 95% CI 1.14–2.57), including MDR VAP (23% vs. 11%, *p* = 0.03), than non-COVID-19 controls	No data	No data	Small sample size, monocentric study
Oliva et al. [18]	No difference in the incidence of ICU-acquired MDR colonization and infection in COVID-19 vs. flu patients	No data	No data	Small sample size, monocentric study
Rouzé et al. [2]	Higher incidence of VA-LTRI and VAP in COVID-19 patients vs. patients with flu or no viral infection, with lower rate of MDR bacteria isolated in COVID-19 patients (23.3%) vs. patients with flu (38.4%) and no viral infection (33.8%)	No data	No data	
Ong et al. [19]	Similar incidence of ICU-acquired infections, including MDR infections in COVID-19 patients vs. controls	No data	No data	Monocentric study, small number of events
Rouyer et al. [20]	Similar rates of ICU-acquired MDR colonization in COVID-19 vs. controls	No data	No data	Incomplete data collection on ICU-acquired MDR infections
Zuglian et al. [21]	Similar frequency of MDR *P. aeruginosa* and *Enterobacteriaceae* among positive samples in COVID-19 patients and controls	No data	No data	
Bahçe et al. [22]	Rates of antibiotic resistance in *K. pneumoniae*, *A. baumanii*, and *P. aeruginosa* in ETA unchanged for most antibiotics. Increased rate of levofloxacin- and ceftazidime-resistant *P. aeruginosa*	No data	No data	Small number of positive ETA samples (lack of power?)
Sathyakamala et al. [23]	Similar rates of MDR Gram-negative bacteria among blood, urine, and respiratory samples in COVID-19 patients vs. controls	No data	No data	No formal statistical comparison, incomplete data collection
Vacheron et al. [24]	Higher incidence of VAP in COVID-19 patients vs. controls (adjusted sHR 1.68, 95% CI 1.45–1.96) with similar frequency of MDR pathogens (except for a lower frequency of MRSA)	No data	No data	
Jeon et al. [25]	ICU-acquired MDR colonization: decreased rates of MRSA, VRE, CRE, and CRAB in COVID-19 vs. non-COVID-19 patients. ICU-acquired MDR infection: decreased rates of MRSA, CRAB, and CRPA, and increased rates of VRE and CRE in COVID-19 vs. non-COVID-19 patients	No data	No data	Registry study with no patient-related data
Vacheron et al. [26]	Higher incidence of VAP in COVID-19 patients (36.9%) than in both control groups (13.4% and 10.6%), with similar proportions of resistant strains	No data	Higher mortality related to VAP in the COVID-19 group (but no data on the impact of COVID-19 status on the association between MDR VAP and outcomes)	
Cogliati Dezza et al. [27]	Lower incidence of ICU-acquired MDR colonization in COVID-19 (47.4%) vs. control patients (81.4%). No significant difference in incidence rate of ICU-acquired BSI with MDR Gram-negative bacteria in COVID-19 vs. control patients.	No data	Among patients with BSI related to MDR Gram-negative bacteria, 30-day mortality higher in COVID-19 patients than in controls (77.8% vs. 21.4%, *p* < 0.0001)	Small sample size, monocentric study
Metan et al. [28]	Similar rates of MDR bacteria in BSI in COVID-19 patients and controls	No data	No data	Small sample size, monocentric study, incomplete reporting
Buetti et al. [29]	Increased incidence of hospital-acquired BSI related to DTR Gram-negative bacteria in COVID-19 vs. controls (19.4% vs. 13%, *p* = 0.017)	No data	Among patients with Gram-negative DTR BSIs, 28-day mortality higher in COVID-19 patients than for controls (83.7% vs. 65.3%, *p* = 0.025)	No adjustment of statistical analysis on patient-related confounders
Lepape et al. [30]	ICU-acquired MDR colonization: higher incidence in COVID-19 patients vs. controls (6.8% vs. 3.7%, *p* < 0.001). ICU-acquired MDR infection: higher incidence of VAP and BSI related to MDR bacteria in COVID-19 patients (MRSA, CRE, ESBL, and ceftazidime-resistant *P. aeruginosa*)	No data	No data	No clear distinction between colonization and infection in some tables
Kinross et al. [31]	Increased incidence of blood cultures positive with *Acinetobacter* spp., including imipenem-resistant *Acinetobacter* spp., during COVID-19 periods vs. pre-pandemic period	No data	No data	Registry study, no patient-related data
Segala et al. [32]	Similar incidence of ICU-acquired infections in COVID-19 patients vs. controls	No data	No data	
Önal et al. [33]	COVID-19 patients had higher incidence of BSI (but similar rates of other ICU-acquired infections) than controls, with similar rates of MDR bacteria	No data	No data	Retrospective, monocentric study
Petrakis et al. [34]	Increased rate of resistance to common antibiotics among isolates of *K. pneumonia*, *A. baumanii*, and *P. aeruginosa* in COVID-19 patients vs. controls	No data	No data	Retrospective, monocentric study
Lee et al. [35]	Increased rate of imipenem-resistant *K. pneumonia* among ICU-acquired infections in COVID-19 vs. non-COVID-19 patients	No data	No data	Registry study with no patient-related data
Chang et al. [36]	No significant change in the incidence of ICU-acquired MDR infections before and after the COVID-19 pandemic	No data	No data	No patient data, no assessment of ICU-acquired non-MDR infection
Kreitmann et al. [37]	COVID-19 patients had higher incidence of ICU-acquired MDR infections (adjusted sHR 2.50, 95% CI 1.90–3.28), but similar incidence of ICU-acquired MDR colonization (adjusted sHR 1.27, 95% CI 0.85–1.88) vs. controls	No data	Occurrence of ICU-acquired MDR colonization and/or infection associated with decreased survival (adjusted cHR 2.61, 95% CI 1.59–4.27) in COVID-19 patients, but not in controls. No impact of the occurrence of ICU-acquired MDR colonization and/or infection on ICU LOS and on duration of IMV in the overall cohort, in COVID-19 patients, and in controls	No assessment of ICU-acquired non-MDR infection
Piantoni et al. [38]	Increased incidence of ICU-acquired BSI related to MDR bacteria, mostly during the period starting after day 15 post-ICU admission (adjusted cHR 4.35, 95% CI 1.58–11.90)	No data	ICU-acquired MDR BSI associated with increased mortality in overall cohort (adjusted HR 1.73, 95% CI 1.0–3.0, with no effect of COVID-19 status). No association between occurrence of ICU-acquired MDR BSI and ICU LOS and IMV duration in overall cohort, in COVID-19 patients, and in controls	No assessment of ICU-acquired non-MDR BSI

BSI, bloodstream infection; CRAB, carbapenem-resistant Acinetobacter baumanii; CRE, carbapenem-resistant Enterobacteriaceae; CRPA, ceftazidim-resistant Pseudomonas aeruginosa; ESBL, extended-spectrum beta-lactamase; ETA, endotracheal aspirate; DTR, difficult-to-treat resistance; ICU, intensive care unit; IMV, invasive mechanical ventilation; LOS, length of stay; MDR, multidrug-resistant; MRSA, methicillin-resistant Staphylococcus aureus; VAP, ventilator-associated pneumonia; VA-LRTI, ventilator-associated lower respiratory tract infection; VRE, vancomycin-resistant enterococcus.

## Data Availability

Not applicable.

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
