# Peer review of "ICU-Acquired Colonization and Infection Related to Multidrug-Resistant Bacteria in COVID-19 Patients: A Narrative Review"

_antibiotics, 2023, doi:10.3390/antibiotics12091464_

Round 1
Reviewer 1 Report
The Authours organized an interesting review about ICu-acquired colonisation and infection in COVID-19 patients. These findings do not appear particularly original but they can be useful in the clinical practice and interesting for the readers.
I suggest to re-evaluate the sections dedicated to the selective decontamination. Selective decontamination is a controversial question with a large literature and I think this review could not represent the most suitable paper for it. I should be better to eliminate this matter.
Author Response
The Authors organized an interesting review about ICu-acquired colonisation and infection in COVID-19 patients. These findings do not appear particularly original but they can be useful in the clinical practice and interesting for the readers.
I suggest to re-evaluate the sections dedicated to the selective decontamination. Selective decontamination is a controversial question with a large literature and I think this review could not represent the most suitable paper for it. I should be better to eliminate this matter.
Although SSD was not evaluated by RCTs in COVID-19 patients, observational data suggest a beneficial effect on ICU-acquired infections in this population. Therefore, we believe that this paragraph might be helpful for readers.
Reviewer 2 Report
Dear authors,
This narrative review is easy to follow and summaries epidemiological data regarding the incidences of ICU-acquired infection/colonization with resistant bacteria among critically-ill COVID-19 patients.
I particularly appreciated your pathophysiology section which is well explained and constructed.
Some comments for the authors to consider as below:
- The section “5. Specific preventive measures” uses only studies not included in the review according to the tables presented in the section “2. Methods”. This secondary objective concerning the preventive measures is possibly not appropriate in this review because not in line with the selection criteria of studies included is this narrative review.
- Figure 1: regarding “Specific preventive measures”, add the IPC measures (such as cohorting, hand hygiene, use of PPE and efforts to decrease patient contacts, like explained lines 219-223) and the nursing staff availability and training (like suggested line 312).
- Tables 1, 2 and 3: studies were presented by ascending order of publication dates; however, the numbering of references did not respect the order of appearance (for example, [27] appears before [26], [30] appears after [31] and [37] …). Please revise the numbering because according to instructions to authors: “References must be numbered in order of appearance in the text”.
- Tables 1, 2 and 3 should appear in results section (like in “3. Epidemiology” with Table 1 after the first paragraph, Table 2 after second paragraph) instead of appearing in methods section (because they are results not methodological elements).
- Line 165: you cited “[2,38]” but the results presented seem only in line with COVAPID study [2].
- Add the reference number [34] in the sentences: “(contrarily to the COVID-BMR study [34])” and “(in line with 173 the COVID-BMR study [34])” (lines 174, 177, 190…).
- The COVID-BMR study is widely promoted through this review (cited many times and in all sections: lines 154, 169, 174, 176, 190, 198, 214, 228, 235, 249, 252, 292…). No other studies included in this review may supplement the results of your previous study to balance and to complete this manuscript? Indeed, some studies included in this review were poorly cited.
- Lines 223-224: you should give some examples to explain what are the “distinct characteristics” harbored by COVID-19 patients.
- Lines 364-365: you should give some examples to explain what are the “Other research directions (…) interesting to explore”.
Some typos and suggested revisions:
- Figure 1: "pathophysiology" instead of "patophysiology".
- Line 177: "REA-REZO surveillance" instead of "REZO-REA surveillance".
- Line 258: "AMR bacteria" instead of “BMR infections” because “BMR” was not defined and not use previously in this paper (except for the study name: "COVID-BMR study").
- Line 329: you may use the acronym “ARDS” instead of “acute respiratory distress syndrome” because this acronym was previously defined and used in the manuscript.
- Lines 228 and 237: “ICU-MDR colonization/infection” instead of “ICU-MDR-col and/or ICU-MDR-inf”
En espérant que mes remarques vous aideront à améliorer votre article.
Cordialement,
Author Response
This narrative review is easy to follow and summaries epidemiological data regarding the incidences of ICU-acquired infection/colonization with resistant bacteria among critically-ill COVID-19 patients.
I particularly appreciated your pathophysiology section which is well explained and constructed.
Some comments for the authors to consider as below:
- The section “5. Specific preventive measures” uses only studies not included in the review according to the tables presented in the section “2. Methods”. This secondary objective concerning the preventive measures is possibly not appropriate in this review because not in line with the selection criteria of studies included is this narrative review.
This is accurate, and to reflect this, we have slightly modified the paragraph on the selection of the studies for the primary endpoints, and added a second paragraph to describe the selection of studies for secondary endpoints.
- Figure 1: regarding “Specific preventive measures”, add the IPC measures (such as cohorting, hand hygiene, use of PPE and efforts to decrease patient contacts, like explained lines 219-223) and the nursing staff availability and training (like suggested line 312).
We have added these measures to the figure.
- Tables 1, 2 and 3: studies were presented by ascending order of publication dates; however, the numbering of references did not respect the order of appearance (for example, [27] appears before [26], [30] appears after [31] and [37] …). Please revise the numbering because according to instructions to authors: “References must be numbered in order of appearance in the text”.
Thanks, we have corrected the order of references.
- Tables 1, 2 and 3 should appear in results section (like in “3. Epidemiology” with Table 1 after the first paragraph, Table 2 after second paragraph) instead of appearing in methods section (because they are results not methodological elements).
We have relocated figures and tables in the manuscript to follow the reviewer’s comment.
- Line 165: you cited “[2,38]” but the results presented seem only in line with COVAPID study [2].
Both references 2 and 38 are from the COVAPID study, and both provide data that were used to estimate the the cumulative incidence of VAP related to MDR bacteria in all 3 groups (11.8% in COVID-19 patients, 11.6% in flu patients and 8.6% in patients with no viral infection), as presented in this review. Thus, we believe it is relevant to include both references.
- Add the reference number [34] in the sentences: “(contrarily to the COVID-BMR study [34])” and “(in line with 173 the COVID-BMR study [34])” (lines 174, 177, 190…).
We have added this reference throughout the manuscript.
- The COVID-BMR study is widely promoted through this review (cited many times and in all sections: lines 154, 169, 174, 176, 190, 198, 214, 228, 235, 249, 252, 292…). No other studies included in this review may supplement the results of your previous study to balance and to complete this manuscript? Indeed, some studies included in this review were poorly cited.
This is a good point, however as presented in the tables and highlighted in the methods and results section, the COVID-BMR study is the only study that fulfilled simultaneously the following criteria: prospective and multicenter design, separate assessment of ICU-acquired colonization and infection with MDR bacteria, statistical adjustment for confounding factors in the association between COVID-19 and ICU-acquired colonization and infection with MDR bacteria. It is mainly because of its strong methodological criteria that we cite this study in the text. We also refer in detail to other retrospective multicenter studies of high methodological and statistical quality (the two studies by Vacheron et al., the COVAPID study, the studies by Lepape et al. et Buetti et al.). Of note, studying the association of COVID-19 with ICU-acquired colonization and infection with MDR bacteria was only a secondary objective of these studies (while it was the main objective of the COVID-BMR study). Finally, the majority of remaining studies are single-center monocentric studies with important methodological limitations, and while we have referred to these studies in the tables, we believe that they provide limited data of sufficient quality to deserve a thorough treatment in the text.
- Lines 223-224: you should give some examples to explain what are the “distinct characteristics” harbored by COVID-19 patients.
We suggest referring to paragraph 4, which offers a detailed discussion on the “distinct characteristics” that could make COVID-19 patients more susceptible to transitioning from colonization to infection with MDR strains (than non-COVID-19 controls), and have added a sentence to guide the reader to this section.
- Lines 364-365: you should give some examples to explain what are the “Other research directions (…) interesting to explore”.
We thank the reviewer for this comment and have added some research directions we believe would be worth exploring.
Some typos and suggested revisions:
- Figure 1: "pathophysiology" instead of "patophysiology".
Thanks, we have corrected the typo.
- Line 177: "REA-REZO surveillance" instead of "REZO-REA surveillance".
Thanks, we have corrected the typo.
- Line 258: "AMR bacteria" instead of “BMR infections” because “BMR” was not defined and not use previously in this paper (except for the study name: "COVID-BMR study").
Thanks, we have corrected the typo.
- Line 329: you may use the acronym “ARDS” instead of “acute respiratory distress syndrome” because this acronym was previously defined and used in the manuscript.
Done.
- Lines 228 and 237: “ICU-MDR colonization/infection” instead of “ICU-MDR-col and/or ICU-MDR-inf”.
Done, thanks.
En espérant que mes remarques vous aideront à améliorer votre article.
Merci pour vos commentaires qui ont effectivement nettement amelioré le manuscript.
Reviewer 3 Report
The manuscript is a narrative review focused on bacterial infections/colonizations in severe COVID-19 patients. The concept is excellent, and the paper is well-written and well-organized. The tables and figures are described and explained effectively. The English language used is clear, with only a few minor spelling errors throughout the paper.
Additionally, I recommend reading the following papers, which could complement this review: 10.3390/idr14030040, 10.3390/idr14030041.
The English language used is clear, with only a few minor spelling errors throughout the paper.
Author Response
The manuscript is a narrative review focused on bacterial infections/colonizations in severe COVID-19 patients. The concept is excellent, and the paper is well-written and well-organized. The tables and figures are described and explained effectively. The English language used is clear, with only a few minor spelling errors throughout the paper.
Additionally, I recommend reading the following papers, which could complement this review: 10.3390/idr14030040, 10.3390/idr14030041.
We thank the reviewer for this positive comment. The first suggested paper relates to the use of Sarilumab among COVID-19 patients, which is not the focus of this review. The second is an interesting study, however because it does not have a control group it does not fulfil our inclusion criteria.
Round 2
Reviewer 1 Report
The Authors did not answer to the queries. As a consequence the evaluations of the revised version of the paper is quite difficult.
I can just provide the same suggestions
Moderate revision
Author Response
We thank the reviewer for his comments.